# Expression of Oocyte Vitellogenesis Receptor Was Regulated by C/EBPα in Developing Follicle of Wanxi White Goose

**DOI:** 10.3390/ani12070874

**Published:** 2022-03-30

**Authors:** Yeye Du, Xingyong Chen, Han Yang, Linghong Sun, Congcong Wei, Wanli Yang, Yutong Zhao, Zhengquan Liu, Zhaoyu Geng

**Affiliations:** 1College of Animal Science and Technology, Anhui Agricultural University, No. 130 Changjiang West Road, Hefei 230036, China; le13966358979@163.com (Y.D.); yanghan687@163.com (H.Y.); sunlinghong1310@163.com (L.S.); 17681322537@163.com (C.W.); 18435131403@163.com (W.Y.); z18255494174@163.com (Y.Z.); zliu13155@gmail.com (Z.L.); 2Anhui Province Key Laboratory of Local Livestock and Poultry Genetic Resource Conservation and Bio-Breeding, Anhui Agricultural University, No. 130 Changjiang West Rd., Hefei 230036, China

**Keywords:** Wanxi White goose, oocyte vitellogenesis receptor, transcription factor, granulosa cell, C/EBPα

## Abstract

**Simple Summary:**

The key factor affecting the development and mature of follicle in poultry is the incorporation of yolk precursor. The aim of this study was to investigate the mechanism of incorporation of yolk precursor into follicle. The correlation was analyzed between reproductive hormones and oocyte vitellogenesis receptor (OVR) expression, and the transcriptional regulatory factors of OVR were predicted, overexpressed and screened by cloning the promoter region, determining the OVR expression and lipid transportation. The expression of OVR was related to the follicle level (primary follicle, developing follicle and mature follicle) (*p* < 0.001), but not the serum hormones concentration (*p* > 0.05). Simultaneously, the expression of OVR was distributed in the granular cell layer and decreased with the maturation of follicles. The mRNA expression level of OVR was significantly increased in goose granulosa cells with C/EBPα and MF3 stimulated. Remarkably, the combined overexpression of C/EBPα and OVR significantly stimulated the transportation of lipid from culture medium to cytoplasm. The C/EBPα is a key transcription factor promoting OVR expression in goose follicle granulosa cells.

**Abstract:**

Yolk precursor was synthesized under regulation of hormone secretion, while the mechanism of its incorporation into follicle is still unknown. The reproductive hormones, oocyte vitellogenesis receptor (OVR) expression at pre-, early-, peak- and ceased-laying period, and localization of Wanxi White goose were determined in this study. The results showed that the concentration of LH was lowest in serum at peak laying period compared to the other periods (*p* < 0.01). Moreover, the concentration of E2 was highest (*p* < 0.01) in serum at early laying period than that of other periods. Moreover, the gene expression level of *OVR* was highest at ceased laying period compared to other periods (*p* = 0.014) and was higher in developing follicles than other follicles (*p* < 0.01). The OVR was distributed in the granular cell layer and decreased with the maturation of follicles. Five transcription factors were predicted in the promoter of *OVR*, then were screened and verified by overexpression in granulosa cells. C/EBPα and MF3 significantly stimulated the expression of *OVR*. The combined overexpression of C/EBPα and OVR significantly stimulated the transportation of lipid from culture medium to cytoplasm. In conclusion, C/EBPα is the key transcription factor promoting OVR expression in goose follicle granulosa cells.

## 1. Introduction

The Wanxi White goose (Anser anser), as a Chinese indigenous breed, is famous worldwide for its strong disease resistance, large body size and excellent goose down quality. However, compared with hen or duck, the egg production performance of goose is lower and the individual egg production varied greatly among groups [1,2]. The egg production of Wanxi White goose is an average of 25 eggs per egg cycle (150 days), which is less than half of that of Italy white geese (55 eggs/cycle) [3]. The Wanxi White goose (Anser anser), a Chinese indigenous breed, is famous worldwide for its quality and quantity of downy feather. However, seasonal laying and strong broodiness result in its low production of eggs [4]. Therefore, it is an urgent problem to improve the laying performance of Wanxi White geese for poultry industry.

The development of follicle is a key factor affecting the production of egg in poultry, which is a complex physiological process of affecting by nutrition, environment and genetics [5,6,7]. Recent evidence indicated that hormones were considered as the crucial factors for the egg laying, which was specifically regulated by synergistic action with follicle-stimulating hormone (FSH) and luteinizing hormone (LH) [8]. The prolactin (PRL) had been confirmed to delay the development of follicle and reduce the production of egg [9,10].

Cholesterol lipids are the key ingredients for follicle development in poultry, which constitute more than 65% of egg yolk. The cholesterol lipids had been confirmed to be synthesized in liver and ovary under the stimulation of estrogen during egg laying in birds [11,12]. The majority of cholesterol was assembled into lipoprotein (such as yolk, very low density lipoprotein) and then was transported to ovary with other lipids/apolipoproteins for the development of follicle by blood circulation. Therefore, the level of blood yolk precursors was closely associated with the maturation of follicle for egg production [13]. The oocyte vitellogenesis receptor (OVR) was generally considered as a key factor regulating the transport of yolk precursors to follicle by endocytosis [14,15] Simultaneously, the mutation of OVR prevented the transportation of yolk precursors from the blood to follicles and inhibited the formulation of mature follicles, which generated decreased production of eggs in hens [16]. Therefore, the OVR expression might be a key factor for regulating follicle development and egg production in poultry. However, the regulatory mechanism of the expression of OVR is still unknown.

As one of the reproductive traits, egg production is affected by follicle maturation and ovulation. Hence, it is of great significance to explore the possible mechanism of OVR in the regulation of follicle development in poultry. The aim of this study was to investigate the mechanism of incorporation of yolk precursor into follicle. The reproductive hormones and oocyte vitellogenesis receptor (OVR) expression and localization of different growth stages of Wanxi White geese (pre-laying, early laying, peak laying and ceased laying periods) were determined in this study, and the transcription factor were predicted and selected for promoting OVR expression in goose follicle granulosa cells to elucidate the mechanisms related to egg production. This study provided the molecular mechanism of yolk formation of Wanxi white geese.

## 2. Materials and Methods

### 2.1. Ethics Statement

The animal experiment was reviewed and approved by the Institutional Animal Care and Use Committee of Anhui Agricultural University (No. SYDW-P20190600601). The experiments were performed in accordance with the Regulations for the Administration of Affairs Concerning Experimental Animals and the Standards for the Administration of Experimental Practices.

### 2.2. Experimental Animals and Sample Collection

Female Wanxi White geese at pre-laying (240 d), early laying (270 d), peak laying (291 d) and ceased laying (364 d) periods were selected for sample collection with six geese per period. Non-anticoagulant and anti-coagulant blood samples were collected from the wing vein of the geese at 10 a.m. after a 12 h fast. Serum was separated for hormones determination and blood lipid isolation for cell culture from the non-anticoagulant blood, while anticoagulant blood was used for total DNA extraction. Geese were euthanized by carbon dioxide anesthesia for tissue sampling. After euthanasia, the entire ovary was taken out immediately. The follicles were divided into three levels according to their diameter: primary follicles (less than 4 mm), developing follicles (5–30 mm) and mature follicles (larger than 35 mm). After washing by PBS, follicle membrane with only the theca and granular layer were collected and stored in nitrogen for RNA extraction.

An additional six geese at 45–50 wks of age during their peak laying period were euthanized by carbon dioxide anesthesia for follicle granular layer separation. The follicles were divided into a hierarchy of F3, F2 and F1, and pre-hierarchy of small yellow follicle (SYF, 5–10 mm) and large white follicle (LWF, 3–5 mm) according to their diameter suggested by Yuan et al. [17] and Wojtysiak et al. [18]. The collected samples were rapidly frozen in liquid nitrogen and stored at −80 °C for RNA extraction.

Another three geese at 45–50 wks of age during their peak laying were euthanized for follicle collection and immunohistochemical analysis. The fresh ovaries and all follicles at various levels were placed in 4% paraformaldehyde fixation fluid.

The last six female animals at 45–50 wks of age during their peak laying period were euthanized for SYF granular layer separation, then were used for the granular cell culture, identification, and transfection.

### 2.3. Analysis of Serum Hormones Concentration

Serum used for hormone determination was isolated from non-anticoagulant blood after delipidation by centrifugation at 3000× *g* for 5 min. The concentration of follicle-stimulating hormone (FSH), luteinizing hormone (LH), prolactin (PRL), progesterone (P4), and estradiol (E2) were determined by radioimmunoassay method by using the radioimmunoassay system (radiometer, xh6080, Xi’an, China) based on the manufacturer’s protocol (North Bio, Beijing, China). All the hormones were detected at Beijing North Biotechnology Research Institute Co, Ltd.

### 2.4. Blood DNA Extraction, PCR Amplification, Sequencing and Transcriptional Regulatory Factors Prediction

Total blood DNA was extracted by genomic DNA extraction kit (Tiangen, Beijing, China). The specific operation steps were performed according to the manufacture’s instructions.

Three pairs of primers (OVR-1~3, Table 1) were designed to amplify the regulatory region of OVR according to the sequence of duck OVR gene sequence (https://www.ncbi.nlm.nih.gov/gene/101796537, accessed on 20 December 2018). The specific fragments were amplified by PCR amplification using the Taq PCR Master Mix (Tiangen, Beijing, China) and verified by agarose gel electrophoresis. The amplified fragments were then recovered and purified according to agarose gel DNA recovery kit (Tiangen, Beijing, China). The purified target fragment was connected to the T vector by using the pMDTM18-T vector Cloning Kit (TAKARA, Dalian, China) and transformed into DH5α competent cells for sequencing. All the steps were performed according to the kit instructions.

Each sequence was spliced and subjected to the OVR transcriptional regulatory factors’ prediction by using the online Promo software (https://www.bitnos.com/transcription-factors-analysis, accessed on 20 November 2019).

### 2.5. RNA Extraction, Reverse Transcription and Real-Time PCR

Total RNA was extracted with the Omega total RNA extraction kit (Omega, Norcross, GA, USA) according to the manufacturer’s instructions. After the integrity of extracted total RNA was confirmed by electrophoresis and concentration determination (Nanodrop 2000, Thermofisher, Waltham, MA, USA), the cDNA was synthesized by using the TRANS Easy Script One-Step gDNA Removal and cDNA Synthesis Super Mix Kit (Quanshijin, Beijing, China) with the reaction system of 20 μL, including 1 μL (1 μg/μL) total RNA/mRNA, 1 μL Anchored Oligo (dT) 18 Primer (0.5 μg/μL), 10 μL 2 × TS Reaction Mix, 1 μL TransScript^®^ RT/RI Enzyme Mix, 1 μL gDNA Remover, then RNase-free Water was added to 20 μL. The PCR reaction conditions were 42 °C for 15 min, 85 °C for 5 s, and 4 °C for reserve.

Quantitative PCR was performed by using Novo Start SYBR qPCR Super Mix Plus (Novoprotein, Shanghai, China) on a cycler (ABI 7500, Thermofisher, Waltham, MA, USA). The reaction system was 20 μL, including 10.0 μL 2 × Novo Start SYBR qPCR Super Mix Plus, 2 μL template, 0.8 μL upstream Primer, 0.8 μL downstream primer, 0.4 μL ROX, and 6 μL RNase-Free ddH2O. The reaction program was 95 °C for 1 min, followed by 40 cycles (95 °C for 20 s, 60 °C for 20 s, and 72 °C for 30 s), and elongated at 72 °C for 5 min. Gene β-actin was set as the internal control. All the primers for quantitative determination of expression were designed by Primer Premier 5 software according to the gene sequence in NCBI (Table 1).

### 2.6. Immunohistochemical Analysis

The tissue section and immunohistochemical procedure were performed according to Wang et al. [19]. The primary antibody of OVR (1:1000, RD Systems, Catalog: AF2258, Minneapolis, MN, USA), and the secondary antibody (1:1000, Servicebio, Catalog: GB23204, Wuhan, China) were used to identify the expression and localization of OVR in each follicle level through the specific light to dark brown staining. The immunohistochemical section was observed under a microscope (IX73, Olympus, Tokyo, Japan) and the image was acquired by using Cell Sens V1.18 64bit software. The staining area of 8–10 images was quantified and staining densitometric analysis was performed using Image J software (V1.8.0.112).

### 2.7. Recombinant Plasmid Construction

The coding sequence of the goose OVR and cDNA of transcription factors were synthesized by Hedgehog Bio Science and Technology Ltd., Shanghai, China. The synthesized cDNA was cloned into the HindⅢ and BamHⅠ polyclonal site of pEGFP-N1 plasmid to generate the pEGFP-N1-C/EBPα (OE-C/EBPα), pEGFP-N1-Cdx-1 (OE-Cdx-1), pEGFP-N1-MafG (OE-MafG) and pEGFP-N1-MF3 (OE-MF3) overexpressed vectors. The OVR coding sequence was cloned into the XhoI and KpnI site to generate the pEGFP-N1-OVR (OE-OVR) overexpressed vector. The pEGFP-N1 vector without cDNA insertion was set as the negative control (NC). All the vectors were stored at −80 °C till further use.

Four sequences of short hairpin RNAs (shRNA) were designed to knock-down OVR expression according to OVR coding sequence and synthesized by Hefei Linmei Biology Technology Co, Ltd. (Hefei, China). Multiple scrambled shRNAs were cloned into the AgeI and EcoRI sites of PTSB-SH-copGFP-2A-PURO plasmid to generate the shRNAs of OVR, named sh-1, sh-2, sh-3, and sh-4, respectively.

### 2.8. Granular Cell Culture, Identification, and Transfection

The granulosa cells were collected from SYF for cell culture. The granular cell culture was based on the description reported by Ji et al. [20] with some modification. The granular cell layer of SYF was first separated and washed with PBS, then the layer was cut into pieces for the digestion by 0.2% collagenase II (Yeasen, Shanghai, China) at 37 °C for 30 min with gentle oscillation every 5 min in a 15 mL tube. The digestion was terminated by M199 complete medium (contain 10% FBS) and filtered with a 200 mesh screen (Suoqiao, Shanghai, China) and centrifuged at 1000× *g* for 5 min at room temperature. The disposit was resuspended in the M199 medium (HyClone, Logan, UT, USA) two times and desolved in 10 mL M199 medium supplemented with 10% FBS (Gibco, New York, NY, USA), 1% penicillin and streptomycin (Gibco, New York, NY, USA).

The granulosa cells that were isolated were cultured in a six-well plate with a slide at the bottom of the plate. After 70–80% confluent cells formed on the slide, the slide was removed and fixed with 4% paraformaldehyde for 30 min, and then washed with PBS for 3 times for 5 min each time. The slide was dried in air before incubation in 70 μL membrane breaking fluid at room temperature for 20 min. After washing with PBS, the slide was then blocked with blocking buffer of 50 μL 3% BSA for 30 min. The slide was incubated overnight with rabbit anti-FSHR antibody (1:1000, Servicebio, Catalog: GB11275-1, Wuhan, China), which was the specific marker of granulosa cells, and then incubated with Cy3-conjugated anti-rabbit IgG (Servicebio, Wuhan, China) for 50 min. The slide was further incubated in DAPI for 10 min and sealed with Antifade Mounting Medium, examined and photographed by Olympus IX73 microscope.

Granulosa cells with a 70–80% confluent monolayer were seeded in a six well culture plate for plasmid transfection. Cells were transfected with 5 μg plasmid using the Exfect Transfection Reagent (Vazyme, Nanjing, China), with the negative control (NC) generated at the same time. After 36 h growth, the transfection efficiency was monitored using an Olympus IX73 fluorescence microsco. Transfection efficiency was quantified using Image J software and presented as % transfected cells/NC cells. The expression level of OVR, C/EBPα, Cdx-1, MafG and MF3 were determined by RT-qPCR after 48 h of transfection.

### 2.9. Lipid Transferring Ability in a Fibroblast Cell Line with Overexpression or Knockdown of OVR Expression

Chicken embryo fibroblast cell lines (DF-1, purchased from Procell Life Science Technology Co, Ltd., Wuhan, China) were cultured in high glucose Dulbecco’s modified Eagle’s medium (HyClone, Logan, UT, USA) with 10% fetal bovine serum and 1% penicillin and streptomycin supplementation. Cell culture was maintained in a 5% CO_2_ humidified atmosphere at 37 °C (Heracell 150i, Thermofisher Scientific, Waltham, MA, USA).

The DF-1 cell was incubated in a six-well plate with slides at the bottom. After grown to approximately 70% confluency, the cells were transfected with OVR overexpressed and/or C/EBPα, MF3 overexpressed and/or shRNA vectors, then incubated in the high glucose DMEM with 400 μg/mL serum lipid. Cells without any operation were set as wild treatment (WT). After 48 h incubation, the slides were fixed, dehydrated, frozen, and stained with Oil-red O to determine the lipid transferring ability in each transfected cell. Cytoplasm lipid was quantified using Image J software and presented as % Oil Red O staining in transfected groups/NC.

### 2.10. Data Analysis

Gene expression level was calculated by using the 2^−∆∆ct^ method. Gene expression in follicles was analyzed by two-way ANOVA with laying period and follicle level as the two main factors. Serum hormone among different laying periods, OVR expression, transfection efficiency, lipid absorption in different transfected cells were analyzed by one-way ANOVA using SAS9.3 software. Duncan’s multiple range test was used to make multiple comparisons. Correlation was used in the analysis of the association between serum hormone and OVR expression. Regression was used in the analysis of OVR expression with follicle level. Data were expressed as mean ± standard deviation. The significance was set at *p* values less than 0.05.

## 3. Results

### 3.1. Serum Hormone Concentration in Wanxi White Goose at Different Laying Period

Hormones were considered as the crucial factors for the development of follicle, which was specifically regulated by synergistic action with follicle-stimulating hormone (FSH) and luteinizing hormone (LH) [4]. As shown in Table 2, the concentrations of LH in serum of Wanxi White goose at pre-laying (5.70 ± 0.53 mIU/mL) and ceased laying (5.17 ± 0.43 mIU/mL) were significantly higher than that of the peak laying period (2.39 ± 0.57 mIU/mL). The concentration of PRL at ceased laying was significantly higher than that of the other period. Simultaneously, the concentration of E2 in serum of Wanxi White goose at the early laying period (588.27 ± 106.61 g/mL) was higher than that of the other periods. However, the concentrations of FSH and P4 in serum of Wanxi White goose were not significantly different (*p* > 0.05).

### 3.2. Expression Level of OVR in Different Follicle Level of Wanxi White Goose at Each Laying Period and Its Correlation with Serum Hormones

As shown in Table 3, the mRNA expression of OVR at ceased laying period was highest than compared to other periods (*p* < 0.05). Moreover, the mRNA expression of OVR was significantly higher in developing follicle than that of the primary and mature follicles (*p* < 0.01). Mature follicles were not found at pre-laying and ceased laying periods. In order to determine whether the level of hormones could affect the development of follicles, the correlations between the level of hormones in serum and the mRNA expression of OVR in different follicles were analyzed in this study. As shown in Table 4, the mRNA expression of OVR was positively correlated with the concentration of FSH in mature follicles (*p* < 0.05). However, for the mRNA expression of OVR in primary and developing follicles, there were no significant correlations with the concentration of FSH in serum (*p* > 0.05). Simultaneously, the mRNA expression of OVR was not significantly correlated with the concentration of LH, PRL, P4 and E2 in serum of all follicles (*p* > 0.05).

### 3.3. Prediction of Transcriptional Regulatory Factor for OVR Expression

The upstream regions of OVR were sequenced and then spliced from −1 to −1058 regulatory sequence (No. MK446725). Transcription factors were predicted online by Promo software. A total of five transcription factors were predicted in the upsteam of the OVR promoter, namely C/EBP alpha [T00107], MafG [T01437], MF3 [T00507], Cdx-1 [T01484], and NF-1 (-like proteins) [T00601], with the binding sites in the regulatory region of AGAAGGAAGTGA, TTAGCTGCTGAAGAAGCAGGC, TGGGG, CAGTAATA, CAGCCAAA, respectively (Figure 1). The expression of OVR might be regulated by these five transcription factors, which was predicted.

### 3.4. Localization of OVR in Different Follicle Levels

The receptor protein localization of OVR expression at different follicle levels of Wanxi White geese (SWF, LWF, SYF, F5, F4, F3, F2 and F1) were determined immunohistochemically in this study. As shown in Figure 2, the receptor protein of OVR was specifically stained in light brown (as indicated with black arrows). Germinal follicle observed in ovary was full of OVR (Figure 2 OV). Notably, the distribution characteristics of receptor protein OVR were dispersedly distributed in follicles and gradually decreased with the development of follicles. As shown in Figure 2, the vitelline was transported by endocytosis with the receptor protein of OVR (as indicated with white arrows) to follicles and then the vitelline was released by exocytosis in the SYF, F5, F4, F3, F2 and F1 follicles. The level of receptor protein OVR was significantly higher in pre-grade follicles than that of the graded follicles (*p* < 0.05). The receptor protein localization of OVR expression was concentrated on the inner surface of the granular cell layer and gradually decreased with the development of follicle (Figure 2 A). Therefore, the granular cells were selected and separated from the follicles to screen for transcription factors used to regulate OVR expression in the follow-up experiment.

### 3.5. The mRNA Expression of OVR and Transcription Factors in the Follicle Granular Layer at Different Follicle Levels

The mRNA expression of OVR and related transcription factors were determined in LWF, SYF and graded follicles of F3, F2 and F1, As shown in Figure 3A, the mRNA expression of OVR was significantly higher in SYF and F3 than that of other follicles (*p* < 0.05). Obviously, the mRNA expression levels of transcription factors with C/EBPα, MafG, MF3 and NF-1 were highest in SYF than that of the other follicles (*p* < 0.05) (Figure 3B–D,F). Simultaneously, the mRNA expression of transcription factors with Cdx-1 was highly expressed in both SYF and LWF (Figure 3E). Therefore, SYF was selected for granulosa cell culture and screened transcription factors were used to regulate OVR expression in follow-up experiments.

### 3.6. Effect of Transcription Factors Overexpression on OVR Gene Expression

FSHR was identified as a specific receptor protein of granular cells. As shown in Figure 4, the specific receptor proteins of FSHR were marked with fluorescence by immune-staining. This result indicated that the separated cells were granular cells and the purity was 100%. The overexpressed or knockdown vectors were individually transfected in to granulosa cells to investigate the effects of transcription factors C/EBPα, MafG, MF3 and Cdx-1 on expression of the OVR gene. As shown in Figure 5A,B, each of the transfected granulosa cells showed similar high effective transfection efficiency according to the expression level of green fluorescent proteins, suggesting that the overexpressed/knockdown vectors were successfully transfected and expressed in granulosa cells. Afterwards, the mRNA expression of OVR in each transfection group was detected by real-time PCR. As compared with the NC group, the mRNA expression levels of OVR were significantly decreased (*p* < 0.05) in granulosa cells of transfected with OE-MafG and OE-Cdx-1 vectors (Figure 5C). Simultaneously, the mRNA expression levels of OVR were significantly increased (*p* < 0.05) in granulosa cells of transfected with OE-OVR, OE-C/EBPα and OE-MF3 vectors. Notably, the mRNA expression of OVR was highest in granulosa cells of transfected with the OE-MF3 vector. The results suggested that the transcription factors of C/EBPα and MF3 were the potential regulatory factors for regulating the expression of OVR.

Moreover, the mRNA expression of OVR was determined in the granulosa cells transfected with knockdown OVR vectors (sh-1, sh-2, sh-3, and sh-4), to investigate the effects of sh-1, sh-2, sh-3 and sh-4 on expression of the OVR gene. As compared to the NC group, the mRNA expression of OVR in granulosa cells transfected with knockdown OVR vectors both were significantly decreased (*p* < 0.05). Notably, the mRNA expression of OVR in granulosa cells transfected with knockdown of sh-3 was lower than that of others. This result suggested that sh-3 could be used as a typical vector for OVR knockout.

### 3.7. Effect of C/EBPα and MF3 on Lipid Transportation in DF-1 Cells

The vectors of OE-C/EBPα and OE-MF3 were transfected individually or together with OE-OVR or sh-3 into DF-1 cells to further investigate the effect of transcription factors with C/EBPα and MF3 on lipid transportation. The transfected DF-1 cells were cultured with medium containing goose serum lipids for monitoring the transportation of lipids. As shown in Figure 6A,B,D,E, the transportation of lipids was significantly increased (*p* < 0.05) in transfected DF-1 cells with the vectors of overexpressed C/EBPα or OVR compared to that of WT and NC groups. In addition, the transportation of lipids was significantly increased (*p* < 0.05) in co-transfected DF-1 cells with the vectors of overexpressed C/EBPα and OVR compared to that of DF-1 cells separately transfected with the vectors of overexpressed C/EBPα or OVR (Figure 6G). However, transfected DF-1 cells with the vectors of overexpressed MF3 could not significantly increase the transportation of lipids (Figure 6C). The results suggested that the transcription factor of C/EBPα could efficiently increase the expression of OVR to promote the transportation of lipids in DF-1 cells (Figure 6J).

## 4. Discussion

Egg laying is considered as reproductive activity of poultry, which is regulated by the hypothalamus-pituitary-gonadal (HPG) axis [21]. The hormones of FSH, LH, PRL, P4 and E2 were demonstrated as key factors for the regulation of follicle ovulation and production of eggs [22]. In this study, the concentrations of LH in serum of Wanxi White goose were significantly decreased at peak laying period, whereas the concentration of LH in serum of Wanxi White goose at the pre-laying period showed the opposite trend. A previous study had demonstrated that the high concentration of LH was associated with initiation of egg production [23]. Therefore, it is reasonable to speculate that the high concentration of LH at the pre-laying period may indicate that a new egg cycle was about to initiate. Moreover, the declining trend of egg production after the peak laying period may be related to the decreased level of LH hormone. A similar trend was also observed in a previous study [24]. The hormone of E2 was identified as a crucial factor for promoting the synthesis/accumulation of plasma lipids, calcium, phosphorus and protein during sexual maturation. Moreover, the hormone of E2 could promote the rapid development of the fallopian tube and stimulate the synthesis/releasing of yolk precursors [25]. In this study, the concentration of E2 in serum of Wanxi White goose at the early laying period was significantly increased, which accelerated the plasma lipids, calcium, phosphorus and protein to be accumulated for satisfying the demands of the development of follicles and fallopian tubes. Moreover, the concentration of E2 in serum was maintained at a high level to promote the synthesis and release of yolk precursors at peak laying [25].

It has been proven that OVR is key regulator for egg production in poultry. The deficient expression of OVR would cause an egg laying disorder in poultry [16]. OVR affected laying performance mainly by regulating the transportation of lipids [15]. At peak laying period, a large amount of lipids was synthesized in the liver and transferred to the ovary through blood transportation for satisfying the demands of the rapid development of follicles. Meanwhile, the OVR was responsible for transporting lipids (mainly yolk VLDL) to the ovary by endocytosing for promoting the development of follicles in the laying period.

The lipids were synthesized in the liver with the stimulation of E2 for promoting the development of follicles at the early laying period [11]. In this study, the mRNA expression of OVR was significantly higher at the ceased laying period than that of other periods, which was probably because the large number of pre-hierarchical follicles was ready for a new round of development. Momentously, the expression level of OVR was not associated with the concentration of serum hormone, but might be regulated by the transcription factor of C/EBPα, which promoted the transportation of lipids in serum by regulating the expression of OVR [26].

Recent evidence indicated that the receptor of OVR was distributed in the large white and F1 follicles of granular cell layer [27,28]. In this study, a large amount receptor of OVR was found in the pre-hierarchical follicles, while distributed less in the hierarchical and F1 follicles by immunohistochemical staining. It was probably because the receptor of OVR had already existed in pre-hierarchical follicles with distribution in the peripheral region of follicles for transporting yolk precursors by endocytosis to facilitate the maturation of follicles for the rapid growth of follicles [6].

Five transcription factors were predicted in the transcriptional region of the OVR gene in this study. The C/EBPα was considered as an important transcription factor for regulating the development of fat deposition [29]. The transcription factor of C/EBPα could increase the concentration of lipids in serum by promoting lipolysis [30]. In this study, we found that the transcription factor of C/EBPα could activate the OVR signaling pathway to promote the transport of blood lipids to follicles for the development of follicles by increasing the expression of OVR. Caudal type homeobox 1 (Cdx-l) was responsible for cell cycle regulation [26,31]. The acetylation protein of MafG might play a vital role in the transcriptional regulation of OVR by enhancing the transcription activity of the NFE2/MafG heterodimer and the binding with DNA [32]. The MF3 was a regulatory gene that participated in the development, growth and lactation reflex of the hypothalamus [33]. Therefore, the regulatory gene of MF3 might be an important factor for regulating the secretion to promote the development and maturation of follicles. The neurofibromatosis type I (NF-1, NF1/CTF) regulated the gene expression through combining with the chromosomal remodeling complex (BAF), which suggested that NF-1 might also be involved in the transcription of OVR [34]. However, the mechanism of regulating the expression of OVR with candidate transcription factors still warrants further exploration. In this study, the overexpressed/knock down vectors of transcription factors were constructed to identify the effective transcription factors for regulating the expression of OVR. The results showed that the expression levels of OVR and each transcription factor were significantly higher in the granular layer of follicles than that of others follicle. Moreover, the distributions of OVR and transcription factors were enriched in the SYF, which suggested the gene expression of OVR was primary in SWF and SYF from the ovary. This suggested that the predicted transcription factors might play key roles in regulating follicle development. In this study, the transcription factors overexpressed for C/EBPα and MF3 could stimulate the expression of OVR in granulosa cells, which suggested the transcription factors of C/EBPα and MF3 were effective for promoting the expression of OVR. Moreover, the overexpressed vectors of C/EBPα and MF3 were used to investigate the transduction of lipids in DF-1 cells with or without co-expressed OVR. The result showed that the transcription factors of C/EBPα were the key transcription facto promoting the expression of OVR in granulosa cells and the transportation of lipids for the development of follicles.

## 5. Conclusions

The expression of OVR was related to the follicle level, but not the serum hormones concentration. Simultaneously, the expression of OVR was distributed in the granular cell layer and decreased with the maturation of follicles. C/EBPα is the key transcription factor for promoting the expression of OVR in goose follicle granulosa cells.

## Figures and Tables

**Figure 1 animals-12-00874-f001:**
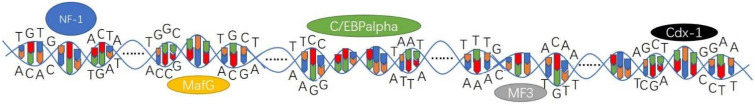
Schematic diagram of binding sites of 5′ transcription factors in the regulatory region of OVR in Wanxi White goose.

**Figure 2 animals-12-00874-f002:**
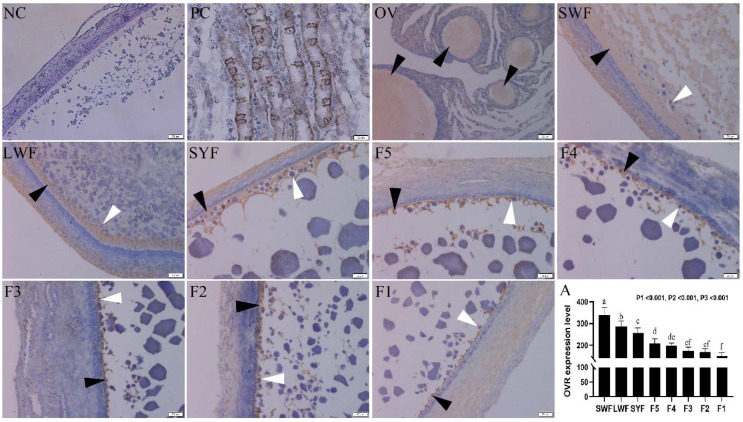
Immunohistochemical staining of oocyte vitellogenesis receptor (OVR) in different levels of follicles of Wanxi White goose. The black arrows indicated the OVR layer and the white arrows indicated the OVR endocytosis of vitelline. NC: negative control; PC: positive control; OV: part of ovary with follicles full of OVR; SWF: Small white follicles; LWF: Large white follicles; SYF: small yellow follicles; F5: F5 follicle; F4: F4 follicle; F3: F3 follicle; F2: F2 follicle; F1: F1 follicle; A: the quantification of OVR in the same area in each level of follicle. Scale bar is 50 µm. a–f: different letter means significant difference among follicle levels (*p* < 0.05).

**Figure 3 animals-12-00874-f003:**
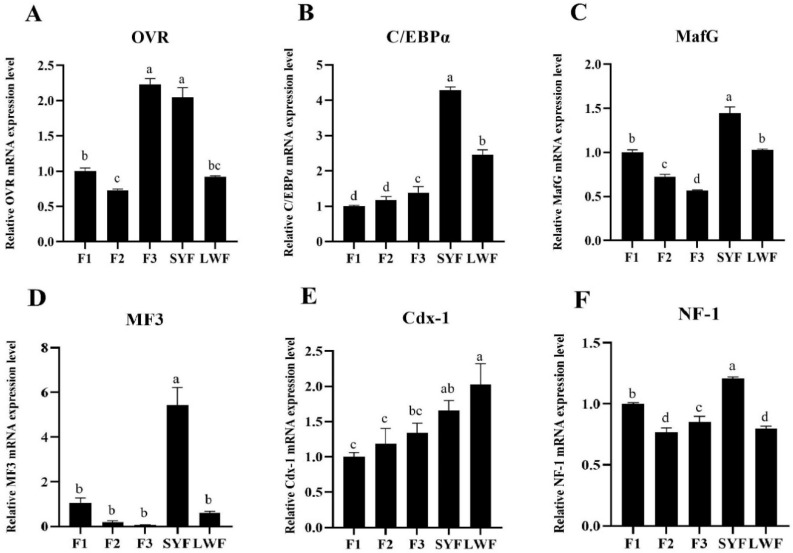
The expression of oocyte vitellogenesis receptor (OVR) and transcription factors at the granular layer in different follicle levels of Wanxi White goose. (**A**): Relative OVR mRNA expression level. (**B**): Relative C/EBPα mRNA expression level. (**C**): Relative MafG mRNA expression level. (**D**): Relative MF3 mRNA expression level. (**E**): Relative Cdx-1 mRNA expression level. (**F**): Relative NF-1 mRNA expression level. a, b, c, d different letters mean significant difference among follicle levels (*p* < 0.05).

**Figure 4 animals-12-00874-f004:**
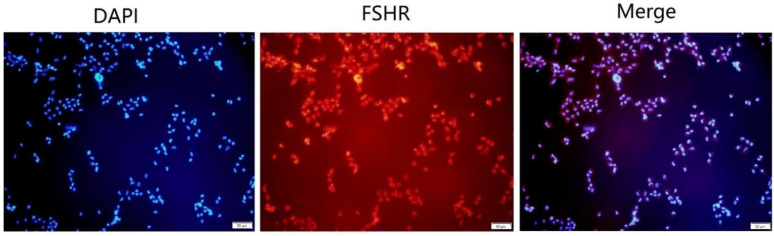
Granulosa cells were identified by immunofluorescence. Granular cell slide was given an immunofluorescent label with FSHR (red, a maker for the granulosa cells). Scale bar is 50 µm.

**Figure 5 animals-12-00874-f005:**
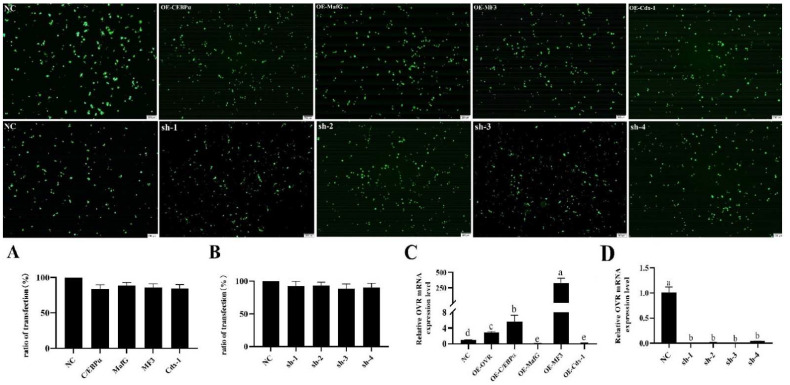
Vector transfection efficiency and OVR expression in each transfected group. NC: negative control; OE-C/EBPα: granulosa cells transfected with overexpressed C/EBPα vector; OE-MafG: granulosa cells transfected with overexpressed MafG vector; OE-MF3: granulosa cells transfected with overexpressed MF3 vector; OE-Cdx-1: granulosa cells transfected with overexpressed Cdx-1 vector. sh-1, sh-2, sh-3 and sh-4 mean granulosa cells transfected with OVR knockdown vectors. (**A**) Granulosa cells transfected with each over-expressed vector. (**B**) Granulosa cells transfected with each OVR knock-down vector. (**C**) Relative OVR expression in each overexpressed vector. (**D**) Relative OVR expression in each OVR knock-down vector.

**Figure 6 animals-12-00874-f006:**
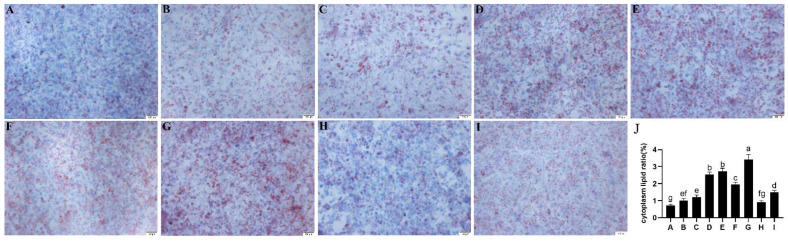
Effect of C/EBPα, MF3 and OVR overexpressed vectors on the transportation of serum lipid. (**A**) DF-1 cells without any performance. (**B**) DF-1 cells transfected with pEGFP-N1 vector without insertion of cDNA. (**C**) DF-1 cells transfected with OE-MF3 vector. (**D**) DF-1 cells transfected with OE-C/EBPα vector. (**E**) DF-1 cells transfected with OE-OVR vector. (**F**) DF-1 cells transfected together with OE-MF3 and OE-OVR vectors. (**G**) DF-1 cells transfected together with OE-C/EBPα and OE-OVR vectors. (**H**) DF-1 cells transfected together with OE-MF3 and sh-3 vectors. (**I**) DF-1 cells transfected together with OE-C/EBPα and sh-3 vectors. (**J**). Cytoplasm lipid ratio (%), a–g: different letter means significant difference (*p* < 0.05).

**Table 1 animals-12-00874-t001:** Primers information for PCR.

Gene	Sequence (5′-3′)	Gene ID	Tm	Product Length
OVR-1	F:GCATGTGCAGCCAAAACTAA R:CCACAAATGAGGGCAGAGAT	101796537	59 °C	379 bp
OVR-2	F:GGGACAGGGCCATACAGTTT R:TCAGTACTCCCCTGCTC ATACA	101796537	60 °C	424 bp
OVR-3	F:TGTATGAGCAGGGGAGTACTGA R:GCGTCCATTACTACACGGGA	101796537	60 °C	492 bp
OVR	F:CCCTCTGAAAAGTAGAGGAGGC R:TGTGTTGGCATTCCAAGGGT	101796537	60 °C	456 bp
CEBPA	F:CTTCTACGAGGTCGATTCCCG R:GATGTCGATGGAGTGCTCGT	110351216	60 °C	172 bp
Cdx-1	F:CCTACGAGTGGATGAGGCG R:GGCATGAATTCCTCCTTGATGGTC	101802595	60 °C	394 bp
MafG	F:AGGGTCCCATCAACAGAGTG R:ATGCCTGCTCTCTTTGTCCT	101793512	60 °C	189 bp
MF3	F:CATGTCAAACATCCCACTGC R:ACCTTGGGCCAATAGGAATC	117001954	58 °C	203 bp
NF-1	F:GCGTGTGCTTGGAAATTTGG R:CCCAGCAAGAAGAGAGACCA	101793953	59 °C	250 bp
β-actin	F:ACACTGTGCCCATCTACGAA R:TCGAAATCCAGGGCGACATA	101800437	60 °C	152 bp

OVR: oocyte vitellogenesis receptor gene; β-actin: beta-actin was set as the reference gene.

**Table 2 animals-12-00874-t002:** Reproductive hormone concentration in Wanxi White goose at different laying periods.

Periods	FSH (mIU/mL)	LH (mIU/mL)	PRL (uIU/mL)	P4 (ng/mL)	E2 (pg/mL)
pre-laying	2.15 ± 0.057	5.70 ± 0.534 ^A^	91.91 ± 16.300 ^B^	0.23 ± 0.101	20.37 ± 5.514 ^C^
early laying	2.71 ± 0.590	4.13 ± 0.929 ^A^	71.97 ± 18.259 ^D^	0.29 ± 0.084	588.27 ± 106.612 ^A^
peak laying	3.03 ± 0.785	2.39 ± 0.568 ^B^	86.4 ± 28.519 ^C^	0.09 ± 0.017	347.53 ± 105.396 ^B^
ceased laying	2.38 ± 0.082	5.17 ± 0.431 ^A^	155.04 ± 54.134 ^A^	0.17 ± 0.031	7.18 ± 1.535 ^C^

^A–D^ differential letters within the same column mean significant difference (*p* < 0.05), the same as below. FSH: follicle-stimulating hormone; LH: luteinizing hormone; PRL: prolactin; P4: progesterone; E2: estradiol.

**Table 3 animals-12-00874-t003:** The mRNA expression of *OVR* in different follicles from Wanxi White goose at different laying periods.

Period	Primary Follicle	Developing Follicle	Mature Follicle
Pre-laying	0.37 ± 0.146 ^D b^	1.36 ± 0.594 ^D a^	/
Early laying	0.66 ± 0.132 ^C b^	3.60 ± 0.715 ^C a^	0.87 ± 0.170 ^A b^
Peak laying	0.86 ± 0.204 ^B b^	4.46 ± 1.821 ^B a^	0.73 ± 0.350 ^A b^
Ceased laying	1.72 ± 0.303 ^A b^	7.30 ± 1.877 ^A a^	/

^A–D^ differential letters within the same column means significant difference (*p* < 0.05). ^a,b^ in the same line means significant difference (*p* < 0.05).

**Table 4 animals-12-00874-t004:** Correlation analysis between serum hormone and OVR expression in different follicle levels of Wanxi White goose.

Follicle Level	FSH	LH	PRL	P4	E2
Primary follicle	0.007	0.014	0.001	0.015	0.006
Developing follicle	0.033	0.027	0.023	0.009	0.043
Mature follicle	0.640 *	0.066	0.045	0.032	0.085

* significant correlation (*p* < 0.05).

## Data Availability

The upstream regions of OVR sequencing data have been deposited to the NCBI (https://www.ncbi.nlm.nih.gov/nuccore/MK446725.1/, accessed on 23 January 2019).

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
