# Peer review of "Expression of Oocyte Vitellogenesis Receptor Was Regulated by C/EBPα in Developing Follicle of Wanxi White Goose"

_animals, 2022, doi:10.3390/ani12070874_

Round 1

Reviewer 1 Report

I have some quitions ans suggestions for the authors

1.- Please state the total number os animal in this work, and the number of animal used in each experiment.

2.- Please the age of geese

3.- Please check the Gne ID in Table 1, some are not the rigth number

4.-Which are the positive and negative control for te immunoflourescence/Histochem experiments?

5- Please mention of anit-FSHR antibody source and the immunofluorescence protocol

6.- How the authors check that the predicted secuence of C/EBP alpha , MafG, MF3, Cdx-1 where in fact the proposed genes? Didi the sequence the amplified fragments?

Reviewer 2 Report

This is an article with excellent information. I have some objections to the presentation of results and the discussion.
The tables are difficult to interpret. They have no units of measurement. The p results are in the middle of two lines, it is not known what they compared with what.
The photographs are excellent, but although the results of each section are shown in the notes to the figures, it is confusing to see how they are expressed in the tables.
When it comes to the discussion, it looks like a literature review of the results and does not show the contribution of the results to a real discussion.
I propose that the investigators review the results and based on the results as a whole make a discussion. There are statements that are questinable, such as how can a single LH measurement in a laying or pre-laying period give conclusive information.
anexoel archivo con notas de preguntas a resolver

Round 2

Reviewer 2 Report

The article improved the discussion and presentation.  I would like to congratulate you